# Unexpected *Amanita phalloides*-Induced Hematotoxicity—Results from a Retrospective Study

**DOI:** 10.3390/toxins16020067

**Published:** 2024-01-29

**Authors:** Miranda Visser, Willemien F. J. Hof, Astrid M. Broek, Amanda van Hoek, Joyce J. de Jong, Daan J. Touw, Bart G. J. Dekkers

**Affiliations:** 1Department of Clinical Pharmacy and Pharmacology, University Medical Center Groningen (UMCG), 9713 GZ Groningen, The Netherlands; m.visser05@umcg.nl (M.V.); w.f.j.hof@umcg.nl (W.F.J.H.); a.m.broek@olvg.nl (A.M.B.); amandavanhoek@home.nl (A.v.H.); joycee-dejong@hotmail.com (J.J.d.J.); d.j.touw@umcg.nl (D.J.T.); 2Department of Pharmaceutical Analysis, Groningen Research Institute of Pharmacy, University of Groningen, 9713 AV Groningen, The Netherlands

**Keywords:** *Amanita phalloides* poisonings, mushroom, liver, kidney, hematological parameters

## Abstract

Introduction: *Amanita phalloides* poisoning is a serious health problem with a mortality rate of 10–40%. Poisonings are characterized by severe liver and kidney toxicity. The effect of *Amanita phalloides* poisonings on hematological parameters has not been systematically evaluated thus far. Methods: Patients with suspected *Amanita phalloides* poisonings were retrospectively selected from the hospital database of the University Medical Center Groningen (UMCG). Medical data—including demographics; liver, kidney, and blood parameters; treatment; and outcomes—were collected. The severity of the poisoning was scored using the poison severity score. Results: Twenty-eight patients were identified who were admitted to the UMCG with suspected *Amanita phalloides* poisoning between 1994 and 2022. A time-dependent decrease was observed for hemoglobin and hematocrit concentrations, leukocytes, and platelets. Six out of twenty-eight patients developed acute liver failure (ALF). Patients with ALF showed a higher increase in liver enzymes, international normalized ratios, and PSS compared to patients without ALF. Conversely, hemoglobin and platelet numbers were decreased even further in these patients. Three out of six patients with ALF died and one patient received a liver transplant. Conclusion: Our study shows that *Amanita phalloides* poisonings may be associated with hematotoxicity in patients. The quantification of hematological parameters is of relevance in intoxicated patients, especially in those with ALF.

## 1. Introduction

*Amanita phalloides* poisoning is a serious health problem, accounting for over 90% of fatalities due to mushrooms [1,2,3]. Most cases of *Amanita phalloides* poisonings occur during the fruiting season, as these mushrooms may be easily mistaken for edible species [4]. *Amanita phalloides* contains three toxins, virotoxins, phallotoxins, and amatoxins. The amatoxins are assumed to be responsible for most of the toxic effects. Amatoxins are thermally stable, highly water-soluble, and resistant to acid degradation, and have a high bioavailability [5,6,7,8]. The inhibition of RNA polymerase II is considered to be the main mechanism of toxicity, leading to the inhibition of protein synthesis and subsequent liver toxicity [7]. Other mechanisms, such as the formation of reactive oxygen species, caspase-3-dependent apoptosis, and the upregulation of tumor necrosis factor-α, may be involved as well [1,5,6,9,10].

In patients, *Amanita phalloides* poisonings start with a latency phase followed by a gastrointestinal phase 12–24 h after the ingestion of the mushrooms [4,11]. During this phase, patients may experience gastrointestinal symptoms, such as abdominal pain, nausea, vomiting, and diarrhea [2,4], which may lead to dehydration, hypoglycemia, acid–base disturbances, and electrolyte disorders [11]. Thereafter, a recovery phase takes place, during which the patients appear to improve; however, the destruction of the liver takes place resulting in an increase in hepatic damage markers, such as alanine aminotransferase (ALT), aspartate aminotransferase (AST), and lactate dehydrogenase (LDH) [2,7,8,11,12]. The hepatorenal phase starts 4–7 days after ingestion and is associated with extensive damage to the liver and other organs, especially the kidneys [6,8,11,13]. Patients may experience liver failure resulting in hemorrhages, convulsions, and multiorgan failure, which may ultimately lead to coma and death [7,8]. By contrast, after the amatoxins have been cleared from the body, patients may fully recover due to the regenerative capacity of the liver [11].

Most therapies are based on the treatment of the symptoms, or on the use of antidotes based on preclinical studies and case reports [5]. The most frequently used antidotes are N-acetylcysteine (NAC), benzylpenicillin (PEN), and silibinin (SIL) [14,15]. No standardized procedures or guidelines for the treatment of *Amanita* poisonings are available [8,13]. Recently, we have studied the effects of antidote therapies on the clinical outcomes of *Amanita* poisonings. Patients received either supportive care alone or in combination with one or more antidotes. Supportive care was defined as standard hospital care, which includes fluid and electrolyte replacement, symptomatic treatment (anti-emetics and anti-diarrheal drugs), corticosteroids, or any other treatment that is not specific for the treatment of *Amanita phalloides* poisonings, such as activated charcoal and dialysis. In that study, the overall survival rate, independent of the treatment administered, of all patients was 84%, while in the group of patients that only received supportive care, the survival rate was 59% [15]. N-acetylcysteine is expected to reduce α-amanitin toxicity through its antioxidant effects, by catching free radicals, and through the synthesis of the antioxidant glutathione, similar to the treatment of acetaminophen poisonings [5,16]. PEN and SIL inhibit the uptake of amatoxins via hepatocytes [5,14]. In addition, SIL shows antioxidant effects and acts as a radical scavenger [7,8]. In our study, we found that the use of SIL or PEN was associated with a clear improvement in survival, while NAC did not appear to improve patient outcomes. No additional effect of combination therapy was observed, while NAC/SIL combination therapy showed positive results comparable to SIL or PEN [15]. In line with these findings, other studies found that NAC, SIL, and PEN alone or in combination were effective in the treatment of *Amanita phalloides* poisonings [17,18,19,20].

Preliminary results from our group suggest that, in addition to liver and kidney toxicity, a reduction in hemoglobin concentration occurs in patients with suspected *Amanita phalloides* poisonings. Based on these observations, we hypothesized that, in addition to hepatotoxicity and nephrotoxicity, *Amanita phalloides* poisonings may be associated with hematological toxicity. Therefore, we retrospectively investigated changes in hematological parameters in patients with suspected *Amanita phalloides* poisonings. In addition, we investigated the outcome of suspected *Amanita phalloides* poisonings in our center.

## 2. Results

### 2.1. Patient Characteristics

In total, 33 patients were identified who were admitted to the University Medical Center Groningen due to a suspected *Amanita phalloides* poisoning between 1994 and 2022. Hepatic transaminases ALT and AST did not increase in five patients, suggesting that these patients were not intoxicated with *Amanita phalloides* but with other mushrooms (not likely cases). These patients were excluded from further analysis. Of the remaining 28 patients with suspected *Amanita phalloides* poisoning, 16 (57%) were male (Table 1).

The median age of these patients was 39 years, and most patients (75%) originated from outside The Netherlands. Patients mostly ingested multiple mushrooms, but the exact number per patient could not be determined. The median time between the ingestion of the mushrooms and hospitalization was 27 h. The median duration of hospitalization was 8 days, and eight patients were admitted into the intensive care unit (ICU) during hospitalization. During hospitalization, none of the cases was proven. Two cases were classified as probable intoxications, while the other patients were classified as possible intoxications (*n* = 26) as described in the Material and Methods section.

Almost all patients received SIL (93%) and NAC (82%), while PEN was administered in only 10 patients (36%). Frequently observed complications were hepatitis, anemia, lactate acidosis, acute kidney injury (AKI), hypokalemia, and infections. Five patients suffered from kidney insufficiency and were dialyzed during hospitalization. One patient underwent Molecular Adsorbent Recirculating System (MARS) therapy as a rescue treatment for hepatic failure.

Most patients (96%) developed acute hepatitis. Six patients (21%) developed acute liver failure (ALF, acute liver damage in combination with hepatic encephalopathy and coagulopathy) of which three patients did not survive the suspected *Amanita phalloides* poisoning. One patient successfully underwent a liver transplant. ALF, as an indicator of serious poisoning, was not associated with higher laboratory or poison severity score (PSS) values [21] at hospitalization. However, ALF was associated with higher peak values of hepatic transaminases (AST and ALT), bilirubin, ammonia, INR, and lactate (Figure 1, Table 2, and Appendix A, *p* < 0.05). PSS ratings were significantly higher in patients with ALF as well (Figure 1D, *p* < 0.001, Appendix A). Kidney damage, indicated by increased creatinine and blood urea nitrogen (BUN) values, was not exclusively associated with the development of ALF (Table 2).

The patients who died were a 65-year-old female (I), a 49-year-old female (II), and a 38-year-old male (III). The patient who received a liver transplantation was a 51-year-old female (IV), and the patient with ALF who survived without the need for a liver transplantation was a 39-year-old male (V). Maximum median ALT and AST values for these patients were 7192 U/L and 5164 U/L, respectively, which is higher than the ALT and AST values for the full dataset (4539 and 3070 U/L, respectively). All patients showed increased maximum ammonia, creatinine, and INR, except for the creatinine of patient V, which remained within the normal range. The PSS was at least 8 for all patients. These patients all received SIL, patients I–IV received NAC, and only patient I received PEN. All patients developed lactate acidosis, and patients I-IV developed anemia and AKI. Patient IV developed bilateral compartment syndrome, multiple infections, and ICU-acquired weakness after a successful liver transplant and received continuous veno-venous hemofiltration. Patients I-III died because of irreversible organ damage. Lastly, as an example of a serious intoxication without dying, patient V recovered without the need for a liver transplant.

### 2.2. Hematological Parameters

Given the fact that anemia was a frequently observed complication, which has not been systematically evaluated, we investigated the impact of the *Amanita phalloides* poisonings on various hematological parameters.

Hemoglobin concentrations significantly decreased in a time-dependent fashion during hospitalization (*p* < 0.001, Figure 2A). In males, hemoglobin was significantly reduced during hospitalization compared to the median value at hospitalization (*p* < 0.01, Appendix A). Although not statistically significant, hemoglobin concentrations decreased below the reference value (7.5 mmol/L) in females as well. No differences were observed at hospitalization in patients who developed ALF compared to those who did not develop ALF. Minimum hemoglobin concentrations were significantly lower in patients who developed ALF (Figure 2D, *p* < 0.01) compared to patients who developed ALF. The decrease in hematocrit concentrations over time is comparable to those for hemoglobin, with significant changes in males (*p* < 0.05) but not in females (Appendix A). As observed for hemoglobin, erythrocytes appeared to decrease, but due to the low number of patients, this did not reach statistical significance. Mean corpuscular volume (MCV), microcytic erythrocytes, reticulocytes, and immature reticulocyte fraction (IRF) did not show significant changes (Appendix A).

Leukocyte concentrations decreased significantly over time (*p* < 0.001, Figure 2B). A similar decrease was observed for neutrophils (Appendix A, *p* < 0.01). At hospitalization, leukocyte and neutrophil cell counts were both above the upper reference value. During hospitalization, counts of both cell types decreased but stayed within the reference limits. No significant differences were observed between patients with ALF compared to those without ALF (Figure 2E). Eosinophil cell counts increased significantly to values above the reference area (*p* < 0.001, Appendix A). For immature granulocytes, lymphocytes, basophils, and monocytes, no significant changes were observed (Appendix A).

Finally, platelet cell count decreased significantly during hospitalization to values around the lower reference limit (*p* < 0.001, Figure 2C). No effect was observed on the platelet large cell ratio (Appendix A). No differences in platelet counts were observed at hospitalization in patients with ALF compared to those without ALF, while minimum platelet counts were clearly lower in patients with ALF than in patients without ALF (*p* < 0.001, Figure 2F).

## 3. Discussion

In this study, we demonstrate that, in addition to hepatotoxicity and nephrotoxicity, *Amanita phalloides* intoxications appear to be associated with hematotoxicity. In our retrospective study, anemia was one of the most frequently observed complications. More detailed analysis demonstrated that hemoglobin, leukocyte, and platelet cell counts were significantly decreased during admission. In line with the literature [2,7,8,11,12], suspected *Amanita phalloides* poisonings were associated with increased liver injury markers and decreased liver function parameters in our patients, including ALT, AST, and INR. As expected, maximum liver damage values were increased even further in patients with ALF. Interestingly, in patients who developed ALF, hemoglobin concentrations and platelet counts also decreased even further compared to patients without ALF. These findings indicate that in addition to markers of liver and kidney damage, the quantification of hematological parameters may be of relevance in poisoned patients, especially in those with ALF.

*Amanita phalloides*-induced hematotoxicity may have an impact on patient care. This may be of particular interest in patients with ALF. In these patients, coagulation, as shown by the increased INR, is already compromised due to the reduced production of coagulation factors by the liver. This, in combination with the reduced number of platelets, may easily lead to bleeding, leading to an even greater reduction in the number of erythrocytes. An example of this potentially vicious circle was seen in our patient who received a liver transplant (patient IV); this patient required multiple blood transfusions after liver transplantation. Other possible treatments could be the administration of blood growth factors or erythropoietic growth factors, like erythropoietin, for a possible better recovery of patients with anemia. Future studies are required to determine if outcomes of *Amanita phalloides* poisonings may be improved via the treatment of the hematotoxicity.

Overall survival in our center was 89%, which is higher than we have previously reported based on the outcomes of case reports and case series [15]. The outcome is, however, similar to others reported earlier [20,22,23,24]. Most patients were treated with the antidote SIL alone or in combination with NAC and/or PEN. In addition, activated charcoal was often administered as supportive care. Our previous study showed no additional effect of combination therapy [15]. Other studies, however, reported favorable results for combination therapy [17,25]. In the present study, most patients received SIL/NAC or SIL/PEN/NAC combination therapy. The outcomes in our center were in line with the use of these antidotes in our previous study [15]. Despite treatment, 11% of the patients did not survive the suspected poisoning with *Amanita phalloides*. Patients often report to the hospital several hours after ingestion. Current treatment is focused on preventing the (re)uptake of amatoxins, while most of these toxins are already taken up by the liver during their first pass. Therefore, it is important to study other treatment options to restore damage to the liver or to reverse the inhibition of RNA polymerase II [7,11].

Several complications were frequently reported. These included complications associated with acute liver injury, such as hepatitis, lactate acidosis, and acute kidney injury [26]. Surprisingly, anemia was also one of the most frequently observed complications. During hospitalization, hemoglobin and hematocrit concentrations decreased time-dependently. Aplastic anemia has been reported as a complication of acute liver failure, but usually develops over a period of weeks or months, suggesting an independent cause [27]. The volume of erythrocytes (MCV and microcytic erythrocytes) did not appear to be affected, indicating that the lower hemoglobin concentrations were not due to smaller erythrocytes [28]. The number of erythrocytes also appeared to decrease during hospitalization, which may be due to reduced synthesis, increased breakdown, or a shorter half-life [29,30]. A significant decrease in platelets was observed as well. In line, we found that platelet numbers also decreased below the reference range for 82% (56 out of 68 patients) of patients with known platelet concentrations in our database of published case reports [15]. The median platelet concentrations in these cases were 87 (39–139) ×10^9^/L, which is in line with the results from the current study. A recent case series, describing the outcome of three cases of *Amanita fuliginea* poisoning, also described thrombocytopenia as a potential complication [31]. As this mushroom also contains amatoxins and virotoxins, the underlying mechanisms may be similar. The decreases in leukocytes and platelets suggest that the hematotoxicity may be due to a direct effect of the toxin on the bone marrow [32]. Hematopoietic stem cells in the bone marrow differentiate into myeloid and lymphoid progenitor cells [33,34]. Myeloid progenitor cells differentiate into platelets, granulocytes, erythrocytes, or monocytes in the blood, while lymphoid progenitor cells differentiate into different subsets of lymphocytes and natural killer cells [33]. Because of the observed effect on erythrocytes and platelets, α-amanitin appears to mainly affect myeloid progenitor cells. Future studies are required to further investigate the effects of α-amanitin, the primary toxin in *Amanita phalloides*, on the proliferation and survival of hematopoietic cells.

At hospitalization, leukocytosis was initially observed, as demonstrated by an increased number of leukocytes, in particular, neutrophils. Leukocytosis may be due to infections, but may also be the result of stress reactions or dehydration during the gastro-intestinal phase of *Amanita phalloides* poisonings [35,36]. During admission, both leukocyte and neutrophil concentrations steadily decreased but stayed within the reference range, suggesting a normalization rather than direct toxicity. The initial burst usually normalizes after 24 h [36]. Surprisingly, even eosinophils appeared to increase. This may be due to the decrease in total leukocytes, but may also be only a relative increase [37].

During the gastrointestinal phase, patients lose water and electrolytes as a result of vomiting and diarrhea. In line, hypokalemia was found to be a frequent complication. Severe dehydration may result in an increase in lactate and a decrease in renal blood flow. To prevent this, patients are supported through the administration of intravenous fluids and electrolyte replacement [3]. Supportive care with fluids may affect blood cell concentrations due to dilution. The effect of the infusion of one liter of normal saline has been shown to significantly decrease hemoglobin, hematocrit, and leukocyte values, even in the absence of hemorrhage [38]. In the present study, hemoglobin and hematocrit steadily decreased over time and did not recover during admission. Moreover, leukocytes and platelets decreased as well, indicating that these effects cannot be explained by initial dilution alone.

The limitations of this study are the small number of patients and the retrospective nature of this study. Although twenty-eight patients could be included in this study, not all measurements were available for every patient. Moreover, only 2 poisonings were confirmed as probable (identified via a mycologist), while 26 poisonings were possible (clinical symptoms resembling an amatoxin poisoning after mushroom ingestion in combination with anamnesis). Even though the number of probable cases is low, patients often could identify the mushroom if a picture was shown. In combination with their symptoms, the increase in liver enzymes, and the exclusion of other causes for liver toxicity, it is likely that these patients were indeed poisoned with *Amanita phalloides*. In future cases, potential *Amanita phalloides* intoxication should be confirmed through identification via a mycologist or via the detection of α-amanitin in blood and/or urine samples. Another potential limitation is the treatment of the patients. Most patients received comparable treatments, which may also have an impact on the hematological parameters we could not control for.

## 4. Conclusions

In conclusion, our results demonstrate that, in addition to hepatotoxicity, *Amanita phalloides* poisonings appear to be associated with hematotoxicity as well. More in vivo data should be gathered through performing full hematological analysis in patients hospitalized with *Amanita phalloides* poisonings, in the future, to determine the role of this form of toxicity on clinical outcomes. In addition, in vivo data on the effects of antidotes on the blood cell count in these patients should be gathered.

## 5. Materials and Methods

### 5.1. Study design and patients

The setup of this investigation was a retrospective, observational study. Patients with *Amanita phalloides* poisonings were retrospectively selected from the hospital database of the University Medical Center Groningen, a tertiary hospital in The Netherlands. For this study, the Medical Ethical Committee of the University Medical Center Groningen (Groningen, The Netherlands) waived the need for written informed consent due to the retrospective nature of the study (reference 2020/073). Patient data were processed according to the Declaration of Helsinki.

### 5.2. Procedures

Patient characteristics (gender, age, country of origin, comorbidities, etc.) were extracted from the electronic health records. Data on the poisonings included the number of mushrooms ingested, start of the symptoms, hospitalization, antidote use, complications, need for liver transplantation, and the outcome of the patient.

Clinical diagnosis of *Amanita phalloides* poisoning was based on the anamnesis, biochemical analyses, and identification of the *Amanita phalloides* mushroom by patients from pictures. For all patients, other causes for liver toxicity, such as acetaminophen poisonings, were excluded. We defined ‘proven cases’ as cases with laboratory confirmation of amatoxins in body fluids; ‘probable cases’ as cases in which mushroom samples were identified via a mycologist, but without laboratory confirmation; ‘possible cases’ as cases where there were neither of these, but the patients showed clinical symptoms that resemble typical amatoxin poisoning after mushroom ingestion; and ‘not likely cases’ as cases in which patients did not show typical amatoxin poisoning symptoms [15].

Laboratory measurements were collected as part of routine care. Liver, kidney, and biochemical parameters (ALT, AST, international normalized ratio (INR), creatinine, ammonia, bilirubin, lactate, creatinine, blood urea nitrogen, LDH, etc.) were assessed by the Department of Laboratory Medicine at the UMCG (ISO15189 accredited) using routine laboratory techniques (Roche, Basel, Switzerland). For these analyses, standard reagents were used, and analyses were performed according to the instructions provided by the manufacturer. INR was maximized at a value of 10. Hematological parameters (hemoglobin, leukocyte, platelets, etc.) were analyzed using a Sysmex XN hematology analyzer using standard reagents and instructions as provided by the manufacturer.

### 5.3. Complications

Complications that occurred frequently (≥5 patients) during hospitalization were collected from the electronic health records, including the seriousness and the outcome. We defined acute liver failure (ALF) as acute liver damage in combination with hepatic encephalopathy and coagulopathy (INR > 1.5) in patients with normal liver function before *Amanita phalloides* poisoning [26]. Acute hepatitis is defined as injury to the liver with elevations in ALT and/or AST >5 times the upper limit of the normal range [39]. We defined lactate acidosis as increased lactate (>2.2 mmol/L), decreased pH (<7.35) and HCO_3_ (<20) [40]. Anemia was defined as hemoglobin concentrations below 7.5 mmol/L (women) or 8.5 mmol/L (men) and hypokalemia was defined as potassium concentrations below 3.5 mmol/L. We defined acute kidney injury (AKI) as increases in serum creatinine by ≥50% within 7 days, serum creatinine by 26.5 μmol/L within 2 days, or oliguria for ≥4 h with a duration < 3 months [41]. Lastly, we defined infections as treatment with an antibiotic other than the antidotes NAC, PEN, and SIL.

### 5.4. Poison Severity Score

The severity of each *Amanita phalloides* poisoning was scored using the poison severity score (PSS) from the European Association of Poisoning Centers and Clinical Toxicologists (EAPCCT) with categories gastro-intestinal tract, respiratory system, nervous system, cardiovascular system, metabolism, liver, and kidneys [21].

### 5.5. Statistical analysis

Blood parameter values were analyzed during the first six days of hospitalization, as the number of patients decreased significantly after this day due to discharge or death. When multiple measurements were performed on the same day, the first available value was used. Statistical analyses were performed in SPSS Statistics software (version 28.0.0.0). Data were presented as median values and ranges. Statistical significance was determined using a non-parametric Mann–Whitney U test (2 groups), or one-way ANOVA followed by a Kruskal–Wallis test (≥3 groups). *p* < 0.05 was considered statistically significant.

## Figures and Tables

**Figure 1 toxins-16-00067-f001:**
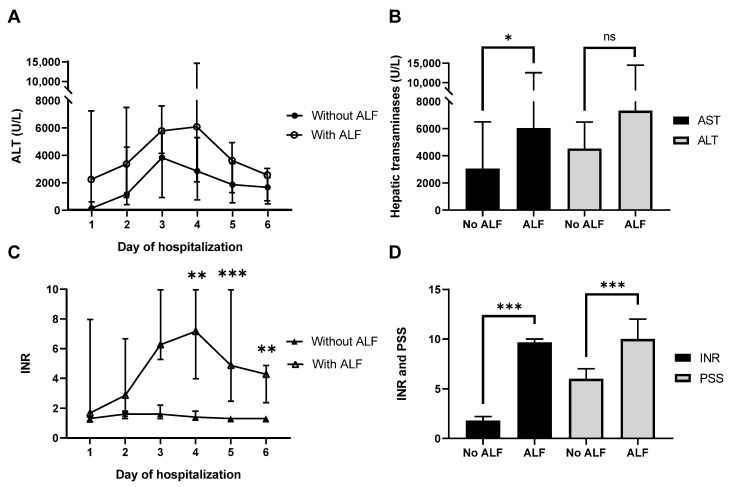
Changes in hepatic transaminases, INR, and poison severity scores (PSSs) in patients with suspected *Amanita phalloides* intoxication. (**A**) ALT concentrations and (**C**) INR over time in patients without or with acute liver failure (ALF). (**B**) Maximum AST and ALT and (**D**) maximum INR and PSS values in patients without and with ALF during hospitalization. Data represent medians and ranges of 22 patients without ALF and 6 patients with ALF. * *p* < 0.05, ** *p* < 0.01, *** *p* < 0.001, ns not statistically significant.

**Figure 2 toxins-16-00067-f002:**
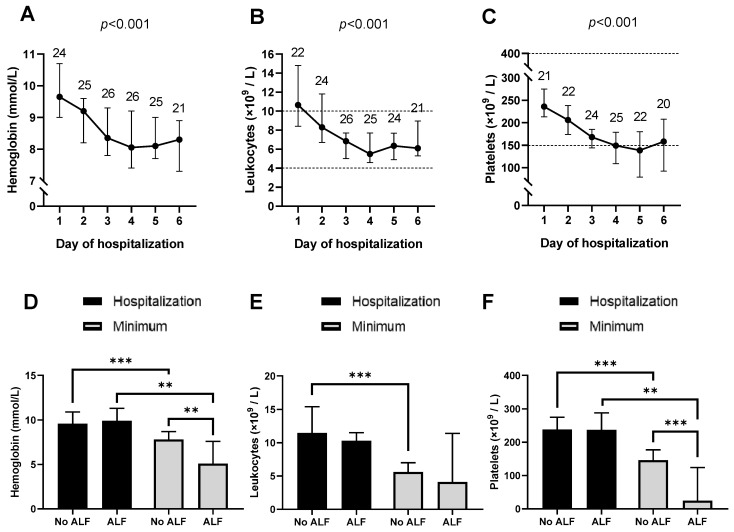
*Amanita phalloides* poisonings appear to reduce hematological parameters in patients. (**A**) Hemoglobin concentrations, (**B**) leukocyte cell counts, and (**C**) platelet cell counts during hospitalization. The number of patients is indicated above the data points. Reference areas are indicated by the dotted lines. *** *p* < 0.001 compared to day 1. (**D**) Concentrations of hemoglobin, (**E**) leukocytes, and (**F**) platelets at hospitalization (black bars) and the minimum values (grey bars) for patients without and with ALF. ** *p* < 0.01, *** *p* < 0.001.

**Table 1 toxins-16-00067-t001:** Patient characteristics.

Characteristic	Total	Patients with Acute Liver Failure (ALF)
Number of patients	28	6
- Male (%)	16 (57)	2 (21)
Age (years)—median (range)	39 (3–72)	44 (3–65)
Origin—number		
- Dutch (%)	7 (25)	1 (17)
- Other (%)	21 (75)	5 (83)
Categorization:		
- Proven (%)	0 (0)	0 (0)
- Probable (%)	2 (6)	0 (0)
- Possible (%)	26 (79)	6 (100)
Hours between ingestion and hospitalization—median (range)	27 (15–96)	28 (17–96)
Days of hospitalization—median (range)	8 (2–64)	15 (2–64)
ICU admission—number (%)	8 (29)	5 (83)
Liver transplantation—number (%)	1 (4)	1 (17)
Survived—number (%)	25 (89)	3 (50)
Antidotes used—number (%)		
- Silibinin	26 (93)	6 (100)
- N-acetylcysteine	23 (82)	5 (83)
- Benzylpenicillin	10 (36)	1 (17)
Complications—number (%)		
- Hepatitis	27 (96)	6 (100)
- Anemia	20 (71)	6 (100)
- Lactate acidosis	10 (36)	6 (100)
- Acute kidney injury	10 (36)	4 (66)
- Acute liver failure	6 (21)	6 (100)
- Hypokalemia	5 (18)	1 (17)
- Infection	5 (18)	1 (17)

**Table 2 toxins-16-00067-t002:** Laboratory values and poison severity score at hospitalization and maximum values during hospitalization.

Laboratory Values	No ALF (*n* = 22)	ALF (*n* = 6)
** At hospitalization **		
Alanine aminotransferase (ALT, U/L)	278 (13–4912)	2197 (62–7192)
Aspartate aminotransferase (AST, U/L)	342 (16–9766)	2246 (83–7285)
Lactate dehydrogenase (LDH, U/L)	285 (132–3192)	3822 (213–6090)
Bilirubin (μmol/L)	14 (4–47)	25 (9–165)
International normalized ratio (INR)	1 (1–2)	2 (1–8)
Ammonia (μmol/L)	30 (7–94)	50 (22–247)
Creatinine (μmol/L)	83 (37–483)	76 (15–286)
Blood urea nitrogen (BUN, mmol/L)	7 (2–29)	6 (2–23)
Lactate (mmol/L)	2 (1–5)	4 (2–10)
** Maximum value during hospitalization **		
Alanine aminotransferase (ALT, U/L)	4539 (158–10,904)	7318 (4414–14,500)
Aspartate aminotransferase (AST, U/L)	3070 (90–10,444)	6045 (3748–12,543) *
Lactate dehydrogenase (LDH, U/L)	2089 (161–9206)	4740 (2280–9527) *
Bilirubin (μmol/L)	55 (15–269)	271 (94–420) **
International normalized ratio (INR)	2 (1–6)	10 (7–10) ***
Ammonia (μmol/L)	57 (30–133)	242 (113–561) ***
Creatinine (μmol/L)	88 (47–1269)	191 (23–286)
Blood urea nitrogen (BUN, mmol/L)	8 (3–29)	12 (6–35)
Lactate (mmol/L)	2 (1–6)	12 (5–22) ***
**Poison Severity Score (PSS)**		
At hospitalization	4 (1–8)	5 (3–10)
Maximum value during hospitalization	6 (2–9)	10 (8–12) ***

Data represent medians (range) of 4–28 patients. * *p* < 0.05, ** *p* < 0.01, *** *p* < 0.001 compared to patients without ALF (no ALF).

## Data Availability

Upon reasonable request, and subject to review, the authors will provide the data that support the findings of this study.

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
