# Peer review of "Unexpected Amanita phalloides-Induced Hematotoxicity—Results from a Retrospective Study"

_toxins, 2024, doi:10.3390/toxins16020067_

Round 1
Reviewer 1 Report
Comments and Suggestions for Authors
In this manuscript, authors studied the “Unexpected Amanita phalloides-induced hematotoxicity – results from a retrospective study”. It is a good piece of research that will work for human welfare. In this manuscript, Authors have studied the effect of Amanita phalloides intoxications on hematological parameters, severe liver and kidney toxicity. Authors have summarized the finding that Amanita phalloides poisoning is associated with decreased hematological parameter values in patients and causes acute liver and kidney toxicity.
Authors should ensure that they have followed the Instructions to Authors of this journal. Overall, the manuscript in its current form is not acceptable for publication in the esteemed “Toxins” journal. Authors require major revisions in the manuscript and can submit the revised manuscript. Some important comments are listed as follows:
1. The scientific name of Amanita phalloides should be italics throughout the manuscript.
2. Abbreviations used first time in the text should be explanatory (Page 1, Line 12).
3. In abstract, authors have selected 33 patients, and have mentioned the outcomes of only 10 patients but what about other patients? Authors should incorporate the data of remaining patients.
4. In introduction, Ist paragraph, authors have cited the same reference many time in same para like 4,5; 5-8; 5-7; 5,8; 6,7. It is recommended that please make the necessary modifications to avoid repetition of the same references.
5. On page 2, Lines 54-55, Authors should mention that on which supportive care, survival rate was 84% & 59%, respectively.
6. In Table 1, proven cases (cases with laboratory confirmation of amatoxins in body fluids) and probable cases (cases in which mushroom samples were identified by a mycologist) were approx 0%, then how we can ensure that the death of three patients was due to Amanita phalloides poisoning. Can we clearly discriminate the patients of Amanita phalloides poisoning only on the basis of clinical symptoms?
7. On page 3, Lines 84-85 Authors have written that During hospitalization, hepatic transaminases ALT and AST did not increase in five patients, but in Fig 1 A clearly indicate an increase in ALT level.
8. In Figure 1 B at No ALF condition, ALT level was above 2000 U/L and AST level was above approx 2500 U/L, and the normal range of ALT & AST is about 40-55 U/L, How authors can report the ALT value 2000U/L under that No ALF condition.
Patients with a marked increase in aminotransferase levels (> 10 times the upper reference limit) typically have acute hepatic injury.
Reference: Giannini EG, Testa R, Savarino V. Liver enzyme alteration: a guide for clinicians. CMAJ. 2005 Feb 1;172(3):367-79. doi: 10.1503/cmaj.1040752. PMID: 15684121; PMCID: PMC545762.
9. The maximaum value of ALT and AST, reported in Figure and Table, does not showing similarity. It is requested that authors should verify the reported values in the Figure and Table.
10. Please cite the references from where methodologies for estimation of parameters were adopted.
11. Please correct the spelling of poisonings throughout the manuscript.
12. In Figure 2D, Hemoglobin content in No ALF and ALF is the same, even in ALF is showing more How?
13. Page 6, Line 162 Authors have written that no differences in platelet numbers were observed at hospitalization but in Figure 2C data shows that there is a decline in the values of platelet numbers during hospitalization.
14. Please correct the table number in manuscript. Both table has the same number.
15. In the discussion section, authors can add a figure related to the mechanism of poisoning (toxicity) in liver and kidney by Amanita phalloides and how drugs (N-acetylcysteine 51 (NAC), benzylpenicillin (PEN), and silibinin (SIL) are reducing the impact of poisoning.
16. Authors have claimed that “In this study we demonstrate for the first time that, in addition to hepatotoxicity and nephrotoxicity, Amanita phalloides intoxications are associated with hematotoxicity” as many other researchers have already reported the Amanita phalloides poisoning. How authors can claim for their work novelty.
Ref: Jander S, Bischoff J. Treatment of Amanita phalloides poisoning: I. Retrospective evaluation of plasmapheresis in 21 patients. Ther Apher. 2000 Aug;4(4):303-7. doi: 10.1046/j.1526-0968.2000.004004303.x.
Author Response
In this manuscript, authors studied the “Unexpected Amanita phalloides-induced hematotoxicity – results from a retrospective study”. It is a good piece of research that will work for human welfare. In this manuscript, Authors have studied the effect of Amanita phalloides intoxications on hematological parameters, severe liver and kidney toxicity. Authors have summarized the finding that Amanita phalloides poisoning is associated with decreased hematological parameter values in patients and causes acute liver and kidney toxicity.
Authors should ensure that they have followed the Instructions to Authors of this journal. Overall, the manuscript in its current form is not acceptable for publication in the esteemed “Toxins” journal. Authors require major revisions in the manuscript and can submit the revised manuscript. Some important comments are listed as follows:
- The scientific name of Amanita phalloides should be italics throughout the manuscript.
This is correct. This mistake occurred during copy-and-pasting all text from the original file to the template file. We have checked the entire manuscript and changed Amanita phalloides over the article into Amanita phalloides, when necessary.
- Abbreviations used first time in the text should be explanatory (Page 1, Line 12).
Thank you for noticing this. We have added “(UMCG)” after the first whole name to make this clear. In addition, we have checked the entire manuscript for abbreviations and checked for explanation at first use.
- In abstract, authors have selected 33 patients, and have mentioned the outcomes of only 10 patients but what about other patients? Authors should incorporate the data of remaining patients.
This may indeed be unclear. We meant to say that six out of a total of 28 patients developed ALF and out of these six patient with ALF, 3 died and 1 received a liver transplant. We have changed the text to make this clear (line 10-16).
- In introduction, Ist paragraph, authors have cited the same reference many time in same para like 4,5; 5-8; 5-7; 5,8; 6,7. It is recommended that please make the necessary modifications to avoid repetition of the same references.
You are right, thank you. We changed the text to avoid repetition of references (line 25-34).
- On page 2, Lines 54-55, Authors should mention that on which supportive care, survival rate was 84% & 59%, respectively.
We defined supportive care and explained a bit more about these patients in the introduction section (line 57-61). Overall survival, independent of the treatment administered, was 84% in this study, while in the group of patients that only received supportive care, survival rate was 59%. We have rephrased this sentence (line 61-63).
- In Table 1, proven cases (cases with laboratory confirmation of amatoxins in body fluids) and probable cases (cases in which mushroom samples were identified by a mycologist) were approx 0%, then how we can ensure that the death of three patients was due to Amanita phalloides poisoning. Can we clearly discriminate the patients of Amanita phalloides poisoning only on the basis of clinical symptoms?
When cases were not proven or probable, diagnosis was based on clinical symptoms, exclusion of other liver toxicity causes and recognition of the mushroom on pictures by patients. We added diagnosis of Amanita phalloides poisonings in the Materials and Methods section (line 302-305). We agree that this is a limitation of our study and included this as such in the discussion section as well (line 276-282).
- On page 3, Lines 84-85 Authors have written that During hospitalization, hepatic transaminases ALT and AST did not increase in five patients, but in Fig 1 A clearly indicate an increase in ALT level.
This was indeed not clearly described. The data in Fig 1A represents the median of all transaminases, in which those 5 patients were initially included. In our revision we excluded these 5 patients as they were classified as not likely, as their admission to our hospital was likely due to other causes than an Amanita phalloides poisoning. We have described the exclusion of these patients in line 86-88 of the revised manuscript.
- In Figure 1 B at No ALF condition, ALT level was above 2000 U/L and AST level was above approx 2500 U/L, and the normal range of ALT & AST is about 40-55 U/L, How authors can report the ALT value 2000U/L under that No ALF condition.
Patients with a marked increase in aminotransferase levels (> 10 times the upper reference limit) typically have acute hepatic injury.
Reference: Giannini EG, Testa R, Savarino V. Liver enzyme alteration: a guide for clinicians. CMAJ. 2005 Feb 1;172(3):367-79. doi: 10.1503/cmaj.1040752. PMID: 15684121; PMCID: PMC545762.
Thank you for this comment. We think there may be an interpretation issue. For each patient, we classified whether there was acute liver failure (ALF) and acute hepatitis. In our manuscript we defined ALF as acute liver damage in combination with hepatic encephalopathy and coagulopathy (INR >1.5) in patients with a normal liver function before an Amanita phalloides poisoning (line 319-322). Acute hepatitis is defined as inflammation of the liver with elevations in ALT and/or AST >5 times the upper limit of normal (line 322-323). Acute hepatitis was observed in nearly all patients. This is now described in the results section (line 107)
- The maximum value of ALT and AST, reported in Figure and Table, does not showing similarity. It is requested that authors should verify the reported values in the Figure and Table.
This was indeed not very presented properly. The data in Figure 1B and 1D shows only maximum values during hospitalization. The data in Figure 1A and 1C demonstrate the changes in ALT and INR values in time. The data in the Table 2 shows values both at hospitalization and the maximum values. The maximum values (second part of the Table) are the same numbers as can be observed in Figure 1B and D, while the hospitalization values correspond with the first day in Figure 1A and C.
We clarified Table 2 by adding “at hospitalization and maximum values” in the title (marked text, line 126) and by underlining both titles in the Table. Moreover, this Table was split over 2 pages in the previous submission. In the submission of the revised manuscript we made sure that the table was presented on 1 page.
- Please cite the references from where methodologies for estimation of parameters were adopted.
We are not entirely sure what you mean by this suggestion as we did not estimate numbers. However, for the definition of proven, probable and possible cases we used the following reference: Tan, J.L.; Stam, J.; van den Berg, A.P.; van Rheenen, P.F.; Dekkers, B.G.J.; Touw, D.J. Amanitin Intoxication: Effects of Therapies on Clinical Outcomes – a Review of 40 Years of Reported Cases. Clin Toxicol 2022, 1–15. This reference has been added (line 310).
- Please correct the spelling of poisonings throughout the manuscript.
We checked the spelling of poisoning and poisonings throughout the manuscript.
- In Figure 2D, Hemoglobin content in No ALF and ALF is the same, even in ALF is showing more How?
Hemoglobin concentrations were similar at hospitalization for patients that did not develop ALF and patients that did develop ALF. During hospitalization the reduction in hemoglobin concentrations was greater in patients that developed ALF compared to patients that did not develop ALF. As a result, minimum values differ in patients with ALF compared to patients without ALF. We have updated the results section accordingly (line 150-155). In addition, we have updated the figure legend to indicate that the black bars present the hemoglobin concentration at hospitalization and the grey bars present the minimum values (line 170).
- Page 6, Line 162 Authors have written that no differences in platelet numbers were observed at hospitalization but in Figure 2C data shows that there is a decline in the values of platelet numbers during hospitalization.
No differences in platelet numbers were observed between patients with and without ALF at hospitalization. During hospitalization, platelet numbers decreased in general. In patients with ALF this decrease was significantly greater compared to decrease in patients without ALF. We changed the order to first describe “at hospitalization” before “minimum values” to clarify this (line 181-184).
- Please correct the table number in manuscript. Both table has the same number.
We apologize. This change occurred when we copied the text into the toxins template. We changed the number into Table 2.
- In the discussion section, authors can add a figure related to the mechanism of poisoning (toxicity) in liver and kidney by Amanita phalloides and how drugs (N-acetylcysteine 51 (NAC), benzylpenicillin (PEN), and silibinin (SIL) are reducing the impact of poisoning.
Thank you for this comment. As this study is more of a descriptive kind, we think the impact of antidotes and mechanisms involved is not an aim of our study. Therefore, we think it is not appropriate to add a figure with the mechanism of poisoning and how antidotes interact.
- Authors have claimed that “In this study we demonstrate for the first time that, in addition to hepatotoxicity and nephrotoxicity, Amanita phalloides intoxications are associated with hematotoxicity” as many other researchers have already reported the Amanita phalloides poisoning. How authors can claim for their work novelty.
Ref: Jander S, Bischoff J. Treatment of Amanita phalloides poisoning: I. Retrospective evaluation of plasmapheresis in 21 patients. Ther Apher. 2000 Aug;4(4):303-7. doi: 10.1046/j.1526-0968.2000.004004303.x.
We agree with the reviewer that outcome of cases with Amanita phalloides intoxications has been described earlier. The novelty of our paper is that hematotoxicity has not been described thus far. During the preparation of the revision, we found another paper describing the thrombocytopenia in patients ingesting Amanita fuliginea. This reference is now included in the manuscript (line 239-242). In addition, we have rephrased the sentences describing novelty.

Reviewer 2 Report
Comments and Suggestions for Authors
This retrospective study describes the findings of 33 cases of suspected Amanita phalloides poisoning with a focus on the hematological parameters. Although retrospective, the study is interesting presenting novel findings of a potential association between poisoning with Amanita phalloides and hematotoxicity. However, some adjustments would improve the quality of the paper and some limitations should be considered and discussed in more detail.
Some specific comments and suggestions:
- One main limitation is that there were no proven and only 2 probable cases. Therefore, descriptions such as “Amanita phalloides induced” or “with Amanita phalloides poisoning” do not seem totally correct and a phrasing such as “suspected Amanita phalloides poisoning” throughout the manuscript would be more accurate.
- Moreover, this limitation should be discussed in more detail, since hematotoxicity after poisoning with other mushroom species (not excluded in this case series) has been described.
- Also regarding this limitation, at the moment unclear why the 5 “not likely” cases were included.
Results section:
- Some of the data presented in the Table are currently also mentioned in the text, better to avoid repetitions.
- “The time between ingestion of the mushroom and hospitalization was 27 hours (15-96 hours)...”: Should be “The median time...”
- Table 1: To be added, what numbers are presented (median (range) probably).
- “In males, hemoglobin was significantly reduced (P<0.05, Supplementary Figure S2).”: Currently not clear in this sentence, reduced compared to what. Through time? Compared to females?
- Perhaps better to avoid using words such as “surprising” in the results section, since results should be presented in a neutral way. You can comment on these aspects in the discussion section (already the case).
Discussion section:
- “Future studies are required to determine if outcomes of Amanita phalloides poisonings may be improved by treatment of the hematotoxicity”: What would be possible treatment options for the hematotoxicity?
- “Our previous study showed that use of SIL or PEN was associated with a clear improvement of survival, while NAC did not appear to improve patient outcome. No additional effect of combination therapy was observed, while NAC/SIL combination therapy showed positive results comparable to SIL or PEN”: Currently also mentioned in the introduction, better to avoid repetitions.
Conclusions section:
- “In conclusion, our results demonstrate that in addition to hepatotoxicity, Amanita phalloides poisoning is also associated with hematotoxicity.”: Same comment as above, since those were suspected cases (and some of them even categorized as “not likely”) better to phrase accordingly, e.g. “might also be...”
Author Response
This retrospective study describes the findings of 33 cases of suspected Amanita phalloides poisoning with a focus on the hematological parameters. Although retrospective, the study is interesting presenting novel findings of a potential association between poisoning with Amanita phalloides and hematotoxicity. However, some adjustments would improve the quality of the paper and some limitations should be considered and discussed in more detail.
Some specific comments and suggestions:
One main limitation is that there were no proven and only 2 probable cases. Therefore, descriptions such as “Amanita phalloides induced” or “with Amanita phalloides poisoning” do not seem totally correct and a phrasing such as “suspected Amanita phalloides poisoning” throughout the manuscript would be more accurate.
We agree with the reviewer. As we are not sure of the poisonings, we changed it to ‘suspected’ Amanita phalloides poisonings for the patients we describe in this study. Owe checked the entire manuscript for these and similar statements we made and changed them accordingly.
Moreover, this limitation should be discussed in more detail, since hematotoxicity after poisoning with other mushroom species (not excluded in this case series) has been described.
Also regarding this limitation, at the moment unclear why the 5 “not likely” cases were included.
This is indeed a limitation of our study. We discussed more in detail how we suspected patients had Amanita phalloides poisoning (line 306-309). In addition, we removed the 5 ‘not likely’ patients from the analysis, please refer to comment 7 of reviewer 1. The text and corresponding data has been adjusted to these changes.
Results section:
Some of the data presented in the Table are currently also mentioned in the text, better to avoid repetitions.
“The time between ingestion of the mushroom and hospitalization was 27 hours (15-96 hours)...”: Should be “The median time...”
Table 2: To be added, what numbers are presented (median (range) probably).
“In males, hemoglobin was significantly reduced (P<0.05, Supplementary Figure S2).”: Currently not clear in this sentence, reduced compared to what. Through time? Compared to females?
Perhaps better to avoid using words such as “surprising” in the results section, since results should be presented in a neutral way. You can comment on these aspects in the discussion section (already the case).
Thank you for these comments. We checked the results section for repetitions. We added the word ‘median’ in the sentence you suggested (line 95). In the initial submission Table 2 was not displayed on a single page, which made it harder to see that beneath this table we have the text: “*data represent median (range) of 4-28 patients.” We made sure this table is now on one page and “at hospitalization” and “maximum values” are underlined to make this difference more clear.
Hemoglobin was significantly reduced in males compared to the median value at hospitalization, we added this in the sentence (line 151-152)
We left the word “surprising” out of our results section to make the text neutral.
Discussion section:
“Future studies are required to determine if outcomes of Amanita phalloides poisonings may be improved by treatment of the hematotoxicity”: What would be possible treatment options for the hematotoxicity?
“Our previous study showed that use of SIL or PEN was associated with a clear improvement of survival, while NAC did not appear to improve patient outcome. No additional effect of combination therapy was observed, while NAC/SIL combination therapy showed positive results comparable to SIL or PEN”: Currently also mentioned in the introduction, better to avoid repetitions.
We have added possible treatment options for hematotoxicity to the discussion section (line 209-219). The sentence has been changed into “Our previous study showed that no additional effect of combination therapy was observed [15].” (line 215-216).
Conclusions section:
“In conclusion, our results demonstrate that in addition to hepatotoxicity, Amanita phalloides poisoning is also associated with hematotoxicity.”: Same comment as above, since those were suspected cases (and some of them even categorized as “not likely”) better to phrase accordingly, e.g. “might also be...”
You are right, we do not know for sure if hematotoxicity is the cause of Amanita phalloides poisonings. We changed this phrase into “may be” or “appears to be” throughout the article (line 18 78, 166, 189, 286).

Round 2
Reviewer 1 Report
Comments and Suggestions for Authors
The revised manuscript “Unexpected Amanita phalloides-induced hematotoxicity – results from a retrospective study” submitted for publication can be accepted for publication in esteemed "Toxins" journal with some minor correction in methodology section. Authors have addressed each queries very well raised by reviewer and have critically modify the manuscript as per requirement. Authors should ensure that they have followed the Instruction to Authors of this journal.
Minor correction:
In Materials and Methods section, Please add fully experimental details as suggested by the learned editor. This may include experimental description regarding estimation of ALT, AST LDH, Bilirubin, Creatinine etc.
Author Response
Laboratory measurements were collected as part of routine care. Liver-, kidney-, and biochemical parameters (ALT, AST, International Normalized Ratio (INR), creatinine, ammonia, bilirubin, lactate, creatinine, blood urea nitrogen, LDH, etc.) were assessed by the department of Laboratory Medicine of the UMCG (ISO15189 accredited) using routine laboratory techniques (Roche, Basel, Switzerland). For these analyses standard reagents were used and analyses were performed according to the instructions provided by the manufacturer. Hematological parameters (Hemoglobin, leukocyte, platelets, etc.) were analyzed using a Sysmex XN hematology analyzer using standard reagents and instructions as provided by the manufacturer. We have reorganized the Materials and methods section and included this information (line 315-323).
Reviewer 2 Report
Comments and Suggestions for Authors
Thank you for addressing all the comments.
Author Response
We thank the reviewer for his/her positive review.